# How Machine Learning Classification Accuracy Changes in a Happiness Dataset with Different Demographic Groups

**Colm Sweeney** [1],*, **Edel Ennis** [1], **Maurice Mulvenna** [2], **Raymond Bond** [2] **and Siobhan O'Neill** [1]

1    School of Psychology, Ulster University, Coleraine BT52 1SA, UK; e.ennis@ulster.ac.uk (E.E.);
     sm.oneill@ulster.ac.uk (S.O.)
2    School of Computing, Ulster University, Jordanstown BT37 0QB, UK; md.mulvenna@ulster.ac.uk (M.M.);
     rb.bond@ulster.ac.uk (R.B.)
*    Correspondence: sweeney-c23@ulster.ac.uk

**Abstract:** This study aims to explore how machine learning classification accuracy changes with different demographic groups. The HappyDB is a dataset that contains over 100,000 happy statements, incorporating demographic information that includes marital status, gender, age, and parenthood status. Using the happiness category field, we test different types of machine learning classifiers to predict what category of happiness the statements belong to, for example, whether they indicate happiness relating to achievement or affection. The tests were initially conducted with three distinct classifiers and the best performing model was the convolutional neural network (CNN) model, which is a deep learning algorithm, achieving an F1 score of 0.897 when used with the complete dataset. This model was then used as the main classifier to further analyze the results and to establish any variety in performance when tested on different demographic groups. We analyzed the results to see if classification accuracy was improved for different demographic groups, and found that the accuracy of prediction within this dataset declined with age, with the exception of the single parent subgroup. The results also showed improved performance for the married and parent subgroups, and lower performances for the non-parent and un-married subgroups, even when investigating a balanced sample.

**Keywords:** machine learning; classification; positive psychology

## 1. Introduction

Positive psychology includes the scientific study of factors that constitute happiness and what people can do, to themselves or others, to affect and improve happiness [1]. Technology can play an important part in understanding what increases positive emotion and in the broader areas of mental health and wellbeing. By using machine learning to automatically classify untagged happiness statements into the specified categories, for example, achievement and affection, a large corpus containing millions of happiness statements could be analyzed, providing summary statistics about what makes the people happy.

Similar work has explored how happiness statements tagged with demographic data can be used to train a machine learning (ML) classifier to predict whether statements were written by a parent or non-parent, a single or married person, a man or woman, or someone who is young or old. Work on the HappyDB by [2] examined whether gender, parenthood, or marital status could be predicted, using a dataset expressing moments of happiness and associated demographic characteristics to indicate whether a statement related to an 18–25-year-old male non-parent or a female parent over 50 years old, for example. Similar work by [3] investigated the use of word embeddings integrated with machine learning classifications to examine gender classification in tweets, finding that word embedding models were more effective when applied to twitter data. A similar algorithm could also be used to classify text from other social media outlets or used to train a chatbot to converse to a user about happiness, but these ideas are beyond the scope of this paper.

Several different models were initially trained using the HappyDB dataset to evaluate the classification accuracy of each model and to investigate how prediction accuracy changed with different types of classification models. The three different models were used to classify the same test data to determine which of them showed the best accuracy. This involved splitting the complete dataset into an 80:20 split, with 80% of the whole dataset being used as training data for the classifier and the remaining 20%, which contained statements that had the category removed (untagged), were used to test the accuracy of the classification model.

Our work is related to the work by [2], which investigated how well generalized algorithms work for different demographic subgroups in the test set. While we saw good overall accuracy scores with the best classification method, we noticed a difference in accuracy scorings when happiness statements were classified for different demographic groups, for example, parental status, age, gender, and marital status. While we understand that 'algorithmic bias' [4] shows that ML algorithms can work well for some groups and less effectively for other groups, it is interesting to determine if an algorithm performs differently with different demographic groups. Where we saw reduced accuracy for certain demographic groups may give an indication that certain groups may use language differently or use more complex language, making it more challenging for the algorithm to classify correctly.

This paper is an extended and improved version of the original paper [5], which was presented at the ICTS4eHealth (IEEE International Conference on ICT Solutions for e-Health) workshop at the ISCC (IEEE Symposium on Computers and Communications (ISCC 2021) conference in September 2021. The main contributions of this extended paper include revised methods, the additional analysis of balanced data where new experiments were undertaken across three age groups and the original results were improved, and comparison with the new balanced samples using new line charts. This was due to one of the age cohorts being notably smaller than the other two age groups. The other contribution involves new assessments being conducted regarding text length and complexity to try to explain the patterns observed. The Flesch Kincaid Grade Level readability formula was used to calculate the complexity of the text, and mean length (per happiness statement) was used to determine the length of texts for different age groups. The single parent group was used for comparison with the other demographic groups.

## 2. Related Works

The HappyDB was created via a crowd-sourcing initiative designed by [6] to assist the study of happiness and the use of machine learning technologies to identify activities that could lead to increased happiness. The HappyDB is a collection of over 100,000 moments of happiness; by answering the question 'What made you happy?', the crowd-sourcing volunteers provided descriptions of happy moments from the immediate past (24 h previously) and in the previous 3 months. The majority of the happiness statements were given by single people (53.4%), whereas 41.9% were married; 41.9% of the respondents were female, and 38.8% had children [7].

Initial research using the HappyDB proved that comprehending different characteristics of happiness is a challenging Natural Language Processing (NLP) problem [6] and, to encourage research in this area, the CL-Aff HappyDB dataset was made available as part of the CL-Aff Shared Task [8]. This is where different teams were encouraged identify a variety of methods to classify happy moments in terms of emotion and content [8]. Various teams were also involved with the 2nd Workshop on Affective Content Analysis@ AAAI (AffCon2019), and numerous papers have been produced to document the analysis and research outcomes from the participating teams [9–18]. There were various differing approaches used in the predictive part of this shared task, and the majority used the deep learning approach of the convolutional neural network (CNN). The primary task was to use a particular classification method for prediction using the labels of Agency and Sociality. The Agency label indicated when the author had control over their happy

moment, and the Social label indicated when other people were involved. Some other researchers investigated different demographic groups identified in the dataset. These included: gender representation; marital status; parenthood distribution; age category; and married vs. single groupings [2,11,19].

Researchers in [2] used the HappyDB data to help answer the question—What makes people happy? Particular attention was given to the language used by males versus females, parents versus non-parents, married versus unmarried people, and between different age groups. They employed three classification algorithms—logistic regression, gradient boosting, and the fastText deep learning algorithm—to predict whether a text was written by a man or a woman, whether the person was married or unmarried, whether the person was a parent or a non-parent, and whether they were young or old. Ref. [9] investigates the use of recommender systems to highlight activities that could improve well-being and happiness for general users and those with low self-esteem. Within this study, participants were encouraged to record three things that made them happy, and the system would recommend activities based on their previous happiness records. Through their research on the HappyDB, Ref. [10] identified social happy moments where the sociality tag texts would be more likely to mention social occasions involving family or friends. They also realized that having an understanding of what makes people happy could have tremendous applications in mental health and governance [10].

The HappyDB consists of brief, written diary entries which correspond to happy moments, and can be viewed as an early form of a happiness diary. This concept was further developed by [11], where participants were required to reconstruct their day in episodes and allocate how happy they felt during these episodes. By using happiness diaries, the individual users can be provided with feedback on the feelings they felt during each activity, or they can be provided with a dashboard overview to show which activities provided different levels of happiness. Finding a positive outlook can engender positive emotions which can then work towards feeling good and hopeful about the future. Ultimately, someone living with depression that has hope that their situation can improve will be more likely to focus on obtaining a future with diminished pain, rather than ending their own life [12].

Having introduced the dataset and any related research, it is useful to point out how classification accuracy can be improved in relation to characteristics of the dataset. In relation to predicting the demographics of an author of text, Ref. [13] found that word usage can vary between demographic groups based on age, gender and education. Previous research has investigated how the author's gender can be predicted using fiction and non-fiction texts [14] and gender and age in blog writing [15]. Similarly, Ref. [16] tried to develop a classifier to predict gender for three distinct age groups (teenagers, people aged in their twenties, and people greater than or equal to thirty).

Some studies investigating language and demographics take a sociolinguistic approach to determine if one particular gender uses longer words than the other, for example. The differences in use of language by gender was investigated by [17], where they analyzed the gender-related use of language in blog texts, and Ref. [18] who identified the predominant use of emoticons by males. Ref. [19] looked at text length and how the age of the author could influence the length of text used. Other approaches integrate Natural Language Processing and supervised machine learning to predict demographics, for example, age or gender, from the text. Different classifiers have been used to predict gender in tweets [19], blog posts [20], and other text fragments that are shorter than tweets [21].

Other dataset characteristics include text complexity, which may affect the accuracy of a text classifier. There are various methods to measure the complexity of a passage of text. The Flesch Kincaid Grade Level method [22] is one of the most popular and widely used readability formulas which assesses the approximate reading grade level of a text document [23].

## 3. Methods

### 3.1. Dataset

The HappyDB was developed by [6], and contains more than 100,000 happy moments. The dataset was categorized using categories inspired by research in positive psychology, including: Achievement; Affection; Bonding; Enjoying the Moment; Exercise; Leisure; and Nature. One of the main categories of happiness is Affection, which corresponds to interactions with family members, loved ones, and pets, whereas bonding statements relate to interactions with friends or colleagues. Achievement is another of the main categories, showing that people are happier when trying to achieve their goals. Table 1 shows the 7 categories of happiness, along with definitions of the categories and examples.

**Table 1.** The categories of happy moments, definitions, and examples.

| Category | Definition | Examples |
|---|---|---|
| Achievement | Employing extra effort to achieve a better-than-expected result. | Finish work. Complete marathon. |
| Affection | Meaningful interaction with family, loved ones, and pets. | Hug. Cuddle. Kiss. |
| Bonding | Meaningful interactions with friends and colleagues. | Have meals with coworkers. Meet friends. |
| Enjoying the Moment | Being aware or reflecting on the present environment. | Have a good time. Mesmerize. |
| Exercise | Intent to exercise or workout. | Run. Bike. Do yoga. Lift weights. |
| Leisure | An activity performed regularly in one's free time for pleasure. | Play games. Watch movies. Bake cookies. |
| Nature | In the open air, in nature. | Garden. Beach. Sunset. Weather. |

### 3.2. Data Science Pipeline

The models were trained on the given training dataset that was derived from a total of approximately 80,000 texts. The data was initially split for testing (20% of 100,535 records) using the code (test = train. sample(frac = 0.2)). The test dataset was printed out as an XLS file, which was plugged in as the test file further in the pipeline.

Then, 80,428 records were used for training the model and 20,107 were used for testing further on in the pipeline. Later in the pipeline, the training data were further split using the code (validation_split = 0.2). Validation training was performed on 64,342 samples and validation testing on the remaining 16,086 samples, for 5 epochs.

The features used were vector representations of words, popularly known as embeddings. Recently, word embedding-based approaches [24,25] have become popular, as they are able to capture the semantics and context of words using the machine learning approach of neural networks. The Word2Vec [24] and GloVe [25] word embedding methods derive a vector representation for each word, one that aims to detect the meaning and the relationships between words by learning how the words co-occur in a corpus. In summary, these methods produce a vector space, where each unique word in the corpus is allocated a corresponding vector in the space.

The first type of classification model used for the first classification task, the naive Bayes classifier, which is known as a simple Bayesian classification algorithm [26], is a method that shows good results for text classification [27] and is a simple probabilistic classifier. Despite its simplicity, the algorithm has been used for text classification in many opinion-mining applications [27,28], and much of its popularity is a result of its simple implementation, low computational cost, and relatively high accuracy. The second classifier involves gradient boosting, which is a popular machine learning algorithm [28] integrating decision trees. Decision trees are a supervised machine learning algorithm that divides the provided training data into smaller and smaller parts in order to identify patterns that can

be used for classification and are suitable for text classification tasks [29]. The last method is the convolutional neural network (CNN), which has been shown to achieve a strong performance in text classification tasks [30]. This involves deep learning using neural network methods and has become quite popular when classifying texts. Of the 11 teams involved in the CL-Aff shared task [8], over half of the teams used some sort of CNN for classification. The method for this research uses a convolutional neural network model using Python's TensorFlow deep learning framework. We also integrated the categorical cross-entropy loss function (see Figure 1 below).

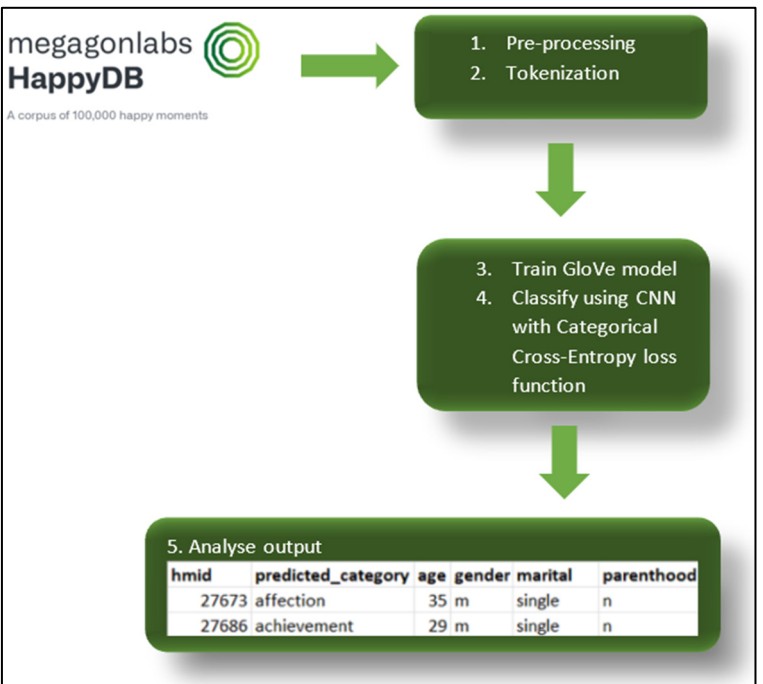

**Figure 1.** The pipeline of the CNN model.

### 3.3. Classification Evaluation

The following four performance measures are commonly used to evaluate classifier applications: classification accuracy; precision; recall; and F1-measure.

### 3.3.1. Classification Accuracy

Classification accuracy is the ratio of correct predictions (True Positive plus True Negative) divided by the total number of predictions made (True Positive plus True Negative plus False Positive plus False Negative).

$$accuracy = \frac{TP + TN}{TP + TN + FP + FN} \tag{1}$$

### 3.3.2. Precision

Precision measures the exactness of a classifier. A higher precision means less false positives, while a lower precision means more false positives. Precision is calculated as: True Positive divided by the sum of True Positive plus False Positive.

$$precision = \frac{TP}{(TP + FP)} \tag{2}$$

### 3.3.3. Classifier Recall

Recall measures the completeness, or sensitivity, of a classifier. Higher recall means less false negatives, while lower recall means more false negatives. Recall is calculated as: True Positive divided by the sum of True Positive plus False Negative.

$$\text{recall} = \frac{\text{TP}}{(\text{TP} + \text{FN})} \tag{3}$$

### 3.3.4. F-Measure Metric

Precision and recall can be combined to produce a single metric known as F-measure, which is the weighted harmonic mean of precision and recall. The best value is 1 and the worst is 0. F1-measure is calculated as below:

$$\text{F1} = 2 \times \frac{(\text{precision} \times \text{recall})}{(\text{precision} + \text{recall})} \tag{4}$$

Positive or negative reference values were used to calculate the precision, recall, and F-measure of the lexicon-based classification approach.

### 3.4. Training Classifiers on Seven Categories of Happiness

Different happiness statements have been categorized into the following categories: Achievement; Affection; Bonding; Enjoying the moment; Exercise; Leisure; and Nature. Ref. [6] chose a set of categories inspired by positive psychology research which also reflected the contents of the HappyDB. They note that the affection category relates to an activity with family members and loved ones, for example, while bonding refers to activities with other people in a person's work or wider social circle. The count of happy statements per category in the HappyDB is shown in Table 2. The greatest proportion (68% of the total) was categorized as either 'affection' or 'achievement', and the things that brought the most happiness involved affection or interactions with family and loved ones. Achievement was a similarly highly placed category, which shows that people can be happier when they feel like they are making progress on their goals.

**Table 2.** Count of records relating to 7 categories.

| Category | Count: Total Dataset | Count: Train | Count: Test |
|---|---|---|---|
| Affection | 34,168 | 27,372 | 6796 |
| Achievement | 33,993 | 27,211 | 6782 |
| Enjoying the Moment | 11,144 | 8905 | 2239 |
| Bonding | 10,727 | 8609 | 2118 |
| Leisure | 7458 | 5922 | 1536 |
| Nature | 1843 | 1456 | 387 |
| Exercise | 1202 | 953 | 249 |
| Total | 100,535 | 80,428 | 20,107 |

### 3.5. Test Dataset Imbalance

A closer analysis of the breakdown of the total responses per age group for the test dataset shows that the 18–25 group had 4817 statements, with the 26–49 age groups having 13,740 statements, and 50 or over group having a total of 1518 statements, which indicates a class imbalance in favor of the 26–49 age group. It could be argued that the group 50 or over provided lower accuracy due to the lower sample spread with the test sample number for that age group equaling 1518, compared to the other age groups with larger sample numbers. This shows a lack of balance in the test dataset and, to reduce this imbalance, new experiments were conducted with the same sample numbers among the groups. This was performed by randomly sampling 1518 samples from both the 26–49 group and the 18–25 group and recreating the graphs from the balanced dataset. The charts generated

with the balanced data are shown along with the original charts with the complete test sample numbers in the results section.

## 4. Results

Three different models were trained to evaluate the performance of each algorithm when classifying statements into one of seven categories including: Achievement; Affection; Bonding; Enjoying the moment; Exercise; Leisure; and Nature). As research by [31] observed that the accuracy of prediction depends on the algorithm selected, we used the three different methods to determine which method produced the best accuracy scoring when predicting which of the seven categories a happiness statement belonged to. This involved splitting the complete dataset into an 80:20 split, with 80% of the whole dataset being used as training data for the classifier and the remaining 20%, which contained statements that had the category removed (untagged), were used to test the accuracy of the classification model. The three methods used include: naive Bayes; gradient boosting: and the convolutional neural network.

The four metrics for evaluating classification models including accuracy, precision, recall, and the F1-measure were used to gauge the performance of the classification algorithms. The naive Bayes approach provided an accuracy score of 82.48%. This was improved with the decision trees (gradient boosting) method, with the accuracy increasing to 86.02%, and the CNN method (with categorical cross-entropy using word embeddings) produced the best accuracy level of 90.83%. This improvement in the three different methods was also reflected in the F1 scoring, which showed increased scores with the naive Bayes (0.854) and gradient boosting (0.883), and for the categorical cross-entropy with embeddings (0.897). A high-level breakdown of the results is shown in Table 3.

**Table 3.** Performance measure per 3 different methods.

|  | Naive Bayes | Gradient Boosting | CNN |
|---|---|---|---|
| Accuracy | 82.48% | 86.02% | 90.83% |
| Recall | 0.85 | 0.87 | 0.89 |
| Precision | 0.86 | 0.89 | 0.90 |
| F1 | 0.854 | 0.883 | 0.897 |

*Displaying Prediction Accuracy for a Classification Task by Predicting for Different Demographic Groups*

The CNN deep learning algorithm produced the best performance for the main classification task involving the seven categories. With the complete test dataset (n = 20,107), it produced an accuracy result of 90.83%, which is shown as the baseline in the following charts.

In Figure 2, various line graphs show the main categories which include gender, marital status and parenthood status, and the relevant accuracy scoring for these demographic groups within three age categories. The chart shows that all of the categories of results performed better for the 18–25 age group, apart from single parents, predicting the 26–49 age groups with less precision, and less again for the 50 or over age group. To further investigate the lack of balance in the test dataset, new experiments were conducted with the same sample numbers among the groups. Having randomly sampled 1518 samples from both the 26–49 group and the 18–25 group, the graphs were recreated from the balanced dataset. The charts generated with the balanced data are shown along with the original charts with the complete test sample numbers; for example, Figure 2 below shows the graph depicting the accuracy results for the complete dataset while Figure 3 shows the same categories for the balanced sample.

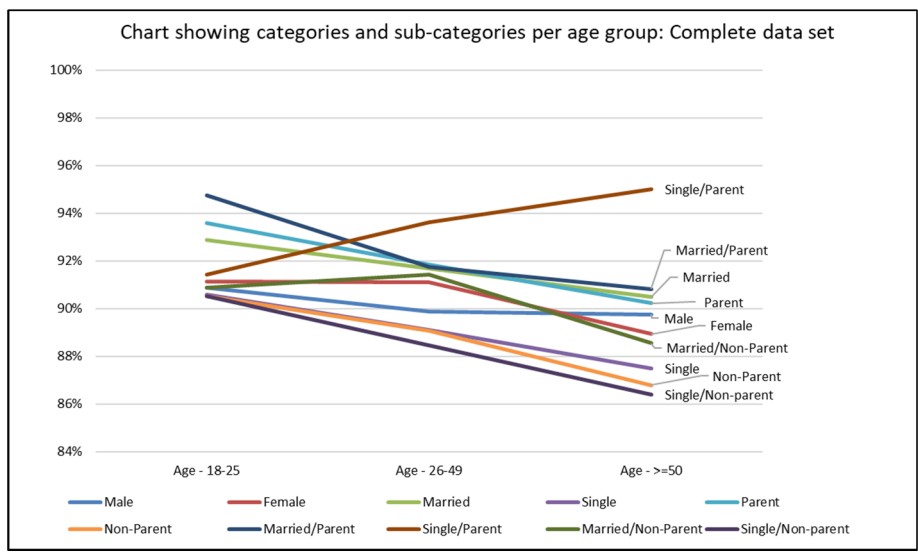

**Figure 2.** Line chart showing accuracy results per age group for gender (male/female), marital status (married/single), and parenthood (parent/non-parent).

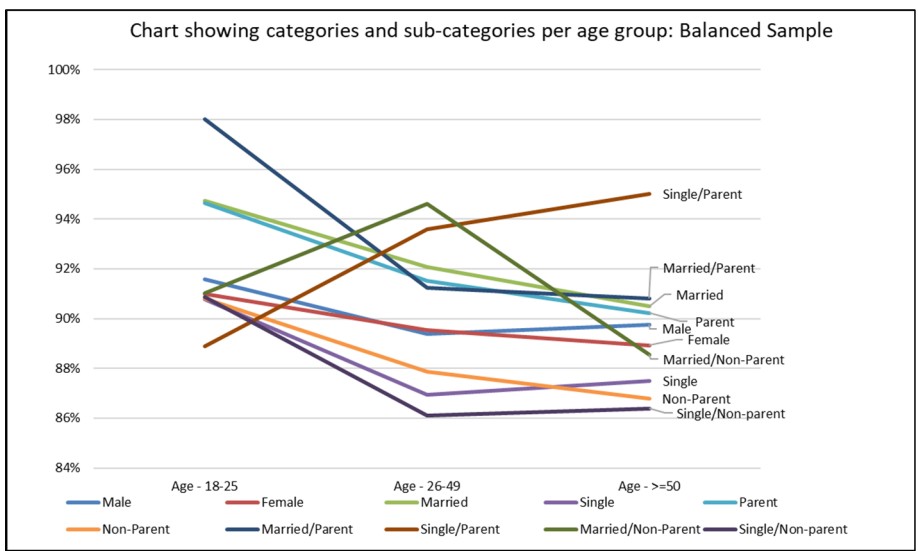

**Figure 3.** Line chart showing accuracy results per age group for gender (male/female), marital status (married/single), and parenthood (parent/non-parent) for the balanced sample.

These show slightly different patterns in the data but, in general, the age group 50 or over was still not predicting as accurately as the 18–25 age group, for example. The single parent group was the one outlier performing differently to all the other groups, with the 18–25 age group showing the lowest accuracy (91.43%), the 26–49 age cohort displaying better performance (93.61%) and the 50 or over age group (n = 40) showing the highest accuracy (95%) in Figure 2. Contrary to this, the married parents demographic scored 98% for the 18–25 age group, as seen in the balanced sample graph (Figure 3).

The accuracy results per age group for marital status, parenthood status, and marital/parenthood for males (Figure 4) showed one demographic subgroup that was performing differently to the other subcategories. This was the single male parent group (n = 2495), where single male parents who belonged to the 50 or over age category (n = 42) showed a 100% accuracy result with this classification system. All the other lines in the graph show prediction patterns similar to Figure 2, apart from married males and male parents, where the result for the age group of 50 or over showed a lower accuracy than the 18–25 age

group. The sub-group showing the lowest accuracy result was the male non-parent aged 50 or over subgroup (n = 156), which generated an accuracy of 85.26%.

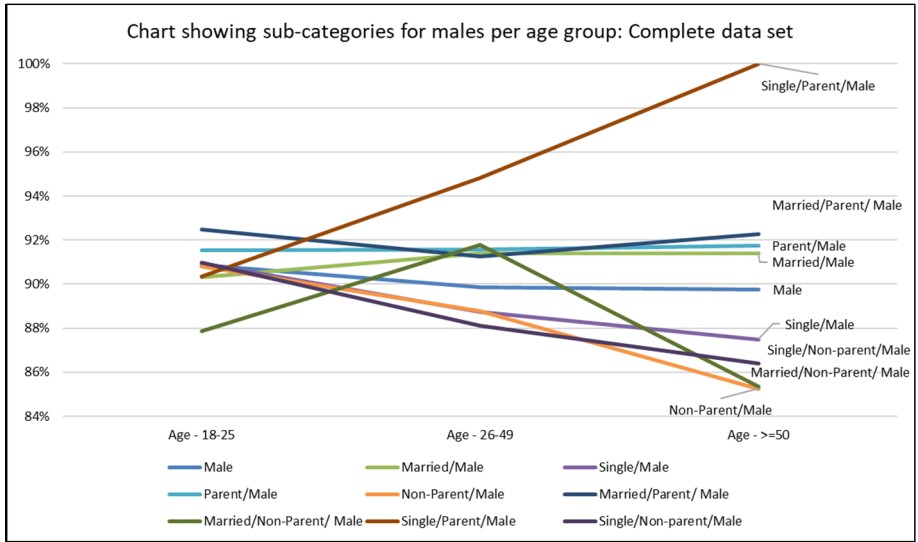

**Figure 4.** Line chart showing accuracy results per age group for marital status (married/single), parenthood (parent/non-parent), and marital/parenthood for gender (male).

The patterns observed in Figure 4 are reflected in Figure 5, where the single male parent group performs contrary to the other groups, with the lowest accuracy result for the 50 or over age group, and 9 out of the 10 sub-demographic groups showing reduced accuracy for the age group of 50 or over when compared to the 18–25 age group.

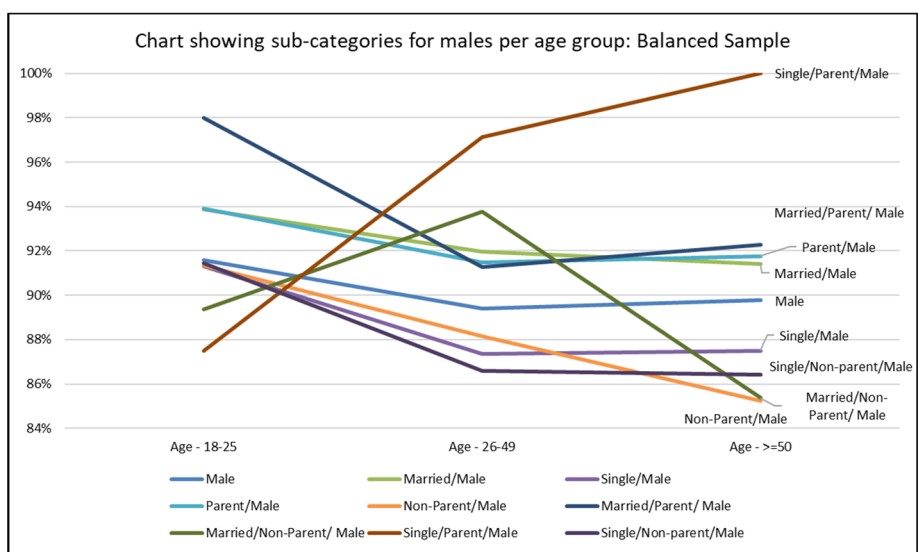

**Figure 5.** Line chart showing accuracy results per age group for marital status (married/single), parenthood (parent/non-parent), and marital/parenthood for gender (male) (balanced sample).

In Figure 6, we can see representation of the gender, marital status and parenthood categories for females. Similar to the previous charts, the one exception is the single parents. All the groups, bar single female parents, show prediction patterns similar to Figure 2, where the percentages for the 50 or over age group demonstrate a lower accuracy result than the 18–25 age group. The highest accuracy is shown for the married female parent aged 18–25 subgroup (n = 140), with 97.14%, and the lowest accuracy result is shown for the single non-parent female aged 50 or over group (n = 169), with 86.39 accuracy. The two

figures below reflect the comparisons seen in the previous charts; Figures 6 and 7 show that accuracy is reduced for all demographic sub-groups when comparing the 18–25 and 50 or over groups, apart from the female single parents who showed accuracy percentages increasing with age.

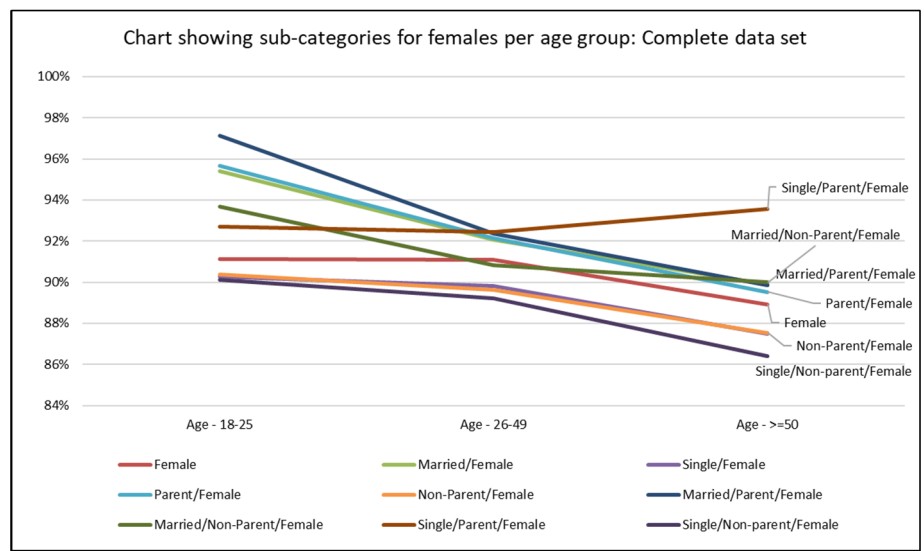

**Figure 6.** Line chart showing accuracy results per age group for marital status (married/single), parenthood (parent/non-parent), and marital/parenthood for gender (female).

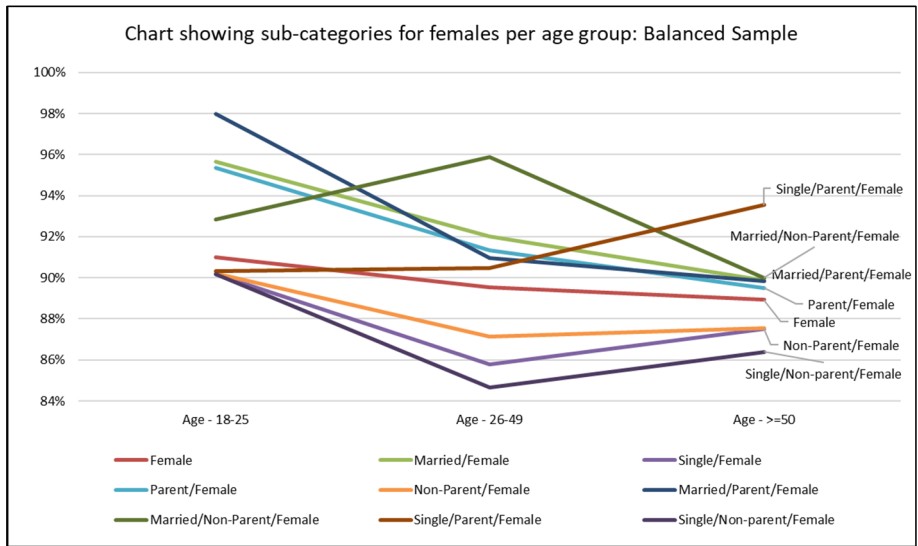

**Figure 7.** Line chart showing accuracy results per age group for marital status (married/single), parenthood (parent/non-parent), and marital/parenthood for gender (female) (balanced sample).

The single parent group performed contrary to all the other groups: the lowest accuracy results were for the 18–25 subgroup, and this increases with the 26–49 and the 50 and over age groups. Some investigation was conducted regarding text length and complexity to try to explain this pattern. The Flesch Kincaid Grade Level readability formula was used to determine the complexity of the text for samples from the complete dataset compared to samples from the single parent only group. The grade level scores calculated were inconclusive, with an increase in text complexity with each older age group for the complete dataset sample, and an increase between the 18–25 group and the over 50s group of single parents. Similarly, text length (which was calculated using the mean length of the complete dataset per age group), showed an increase with each older age group for the complete

dataset (median word length per statement for the 18–25 age group = 13; 26–49 age group = 14; over 50 group = 15), but not for the single parent age groups, which showed the same median text length for the 18–25 and 26–49 age groups (13) and greater for the over 50s group (15). This did not reflect what was expected, i.e., an increase in text length should improve the classification task as the longer text will provide more feature words and lead to the improved performance of the classifier [32].

Regarding other demographics, the results for the complete dataset showed that parents and married people demonstrated higher predicted accuracies than the baseline (90.83%) for all age groups apart from the 50 or over group. Single male parents who were aged 50 or over showed a 100% accuracy score, which may have to be disregarded due to the sample size. The next highest scoring group included the married female parents who were aged 18–25, which showed an accuracy result of 97.14%. On the other end of the scale, the cohort that showed the lowest accuracy was the male non-parent demographic aged 50 or over. This group calculated an accuracy of 85.26%. Married people presented higher accuracy percentages than single people but, unusually, single parents showed higher classification accuracies than the baseline for all subgroups per age. Single non-parent subgroups presented lower percentages than the baseline, with single/non-parents aged 50 or greater showing the poorest accuracy result (86.40%) of all the cohorts. This demonstrates that the algorithm shows better performance for married and parent subgroups, and does not perform so well for non-parent or single subgroups.

A breakdown of all the results, per demographic groups are shown in Table 4, for the complete dataset. This table also includes an indication of the number per each demographic group, in the column with the 'n' heading.

**Table 4.** Table showing accuracy results for demographic groups with numbers per cohort.

| Group Name | All Ages | 18–25 | 26–49 | >=50 |
|---|---|---|---|---|
| Male | 90.13% | 90.88% | 89.86% | 89.76% |
| n | 11,555 | 3017 | 8011 | 508 |
| Female | 90.83% | 91.141% | 91.09% | 88.93% |
| n | 8422 | 1761 | 5645 | 1003 |
| Married | 91.62% | 92.87% | 91.68% | 90.48% |
| n | 8351 | 561 | 6886 | 893 |
| Single | 89.63% | 90.58% | 89.09% | 87.50% |
| n | 10,763 | 4236 | 6196 | 312 |
| Parent | 91.72% | 93.58% | 91.83% | 90.23% |
| n | 7981 | 514 | 6389 | 1064 |
| Non-Parent | 89.50% | 90.54% | 89.07% | 86.78% |
| n | 12,119 | 4301 | 7346 | 454 |
| Married/Male | 91.33% | 90.32% | 91.40% | 91.41% |
| n | 4174 | 279 | 3569 | 326 |
| Single/Male | 89.57% | 90.94% | 88.74% | 87.50% |
| n | 7050 | 2737 | 4182 | 112 |
| Married/Female | 91.96% | 95.39% | 92.06% | 89.88% |
| n | 4130 | 282 | 3274 | 563 |
| Single/Female | 89.88% | 90.27% | 89.83% | 87.50% |
| n | 3637 | 1460 | 1977 | 200 |
| Married/Parent | 91.76% | 94.76% | 91.76% | 90.81% |
| n | 6288 | 286 | 5230 | 762 |
| Single/Parent | 93.23% | 91.43% | 93.61% | 95.00% |
| n | 1019 | 210 | 767 | 40 |
| Parent/Male | 91.60% | 91.54% | 91.59% | 91.76% |
| n | 3680 | 260 | 3066 | 352 |
| Non-Parent/Male | 89.43% | 90.82% | 88.79% | 85.26% |
| n | 7871 | 2755 | 4943 | 156 |

**Table 4.** *Cont.*

| Group Name | All Ages | 18–25 | 26–49 | >=50 |
|---|---|---|---|---|
| Parent/Female | 91.88% | 95.67% | 92.13% | 89.52% |
| n | 4239 | 254 | 3267 | 706 |
| Non-Parent/Female | 89.76% | 90.38% | 89.64% | 87.54% |
| n | 4180 | 1507 | 2375 | 297 |
| Married/Non-Parent | 91.16% | 90.88% | 91.41% | 88.55% |
| n | 2059 | 274 | 1653 | 131 |
| Single/Non-parent | 89.25% | 90.53% | 88.45% | 86.40% |
| n | 9741 | 4025 | 5427 | 272 |
| Married/Parent/ Male | 91.41% | 92.47% | 91.25% | 92.28% |
| n | 2991 | 146 | 2560 | 285 |
| Married/Non-Parent/ Male | 91.11% | 87.88% | 91.77% | 85.37% |
| n | 1181 | 132 | 1008 | 41 |
| Married/Parent/Female | 92.16% | 97.14% | 92.35% | 89.85% |
| n | 3251 | 140 | 2628 | 473 |
| Married/Non-Parent/Female | 91.22% | 93.66% | 90.84% | 90.00% |
| n | 877 | 142 | 644 | 90 |
| Single/Parent/Male | 93.92% | 90.35% | 94.81% | 100.00% |
| n | 510 | 114 | 385 | 9 |
| Single/Parent/Female | 92.56% | 92.71% | 92.43% | 93.55% |
| n | 497 | 96 | 370 | 31 |
| Single/Non-parent/Male | 89.23% | 90.96% | 88.12% | 86.41% |
| n | 6538 | 2622 | 3796 | 103 |
| Single/Non-parent/Female | 89.46% | 90.10% | 89.23% | 86.39% |
| n | 3139 | 1364 | 1606 | 169 |

## 5. Discussion

This work introduces an experiment that was carried out to ascertain if trained machine learning classifiers (using happiness statements) were able to classify unlabeled happiness statements into one of seven different categories. The main contributions of this extended paper include additional analysis of the balanced data and new assessments regarding text length and complexity to try to explain the patterns observed. The Flesch Kincaid Grade Level readability formula was used to calculate the complexity of the text, mean length (per happiness statement) was used to determine the length of texts for the different age groups, and the single parent group was used for comparison with the other demographic groups. Even though the complexity scores calculated were inconclusive and the median text length analysis did not reflect what was expected (i.e., an increase in text length should improve the classification task), the author feels that it has been important to try to explain why the results showed lower accuracies for the single parent sub-group.

Once the accuracy of these results was established for the different demographic categories, including age, gender, and marital and parenthood status, it was then possible to work out to what use these insights could be put to. A smart speaker or chatbot system could use this type of algorithm to ensure the appropriate category of chat during a conversation and also to enable context memory during the interaction. If the conversation was based on leisure, for example, the algorithm could offer dialogue and postings to items relating to leisure. The algorithm could also remind the user about activities that have brought them happiness in the past. This concept is similar to [9], but would be more beneficial as it would have more relevance to a particular demographic group. Older people, for example, tend to have a higher click rate than younger people [33], which could offset the lower accuracies produced by this demographic group. Additionally, if previous happy events were written into a happiness diary, this type of algorithm could use this information to provide suggestions of an activity that has brought happiness in the past.

The course of a conversation will depend on the category being discussed, as will the context of the conversation. For instance, getting the context wrong can be off-putting for the user. The chatbot's response to a line in a conversation relating to achievement, for

example, could be to congratulate the user, but the same response would not be acceptable when the user is enjoying a good movie. There are many implications for the development and deployment of affective computing systems in the area of mental health care and, in particular, when dealing with depression. The developers of these applications should remember that the design of such systems could have life and death consequences, as incorrect use of language or responses could lead a person into a darker place than they were originally.

## 6. Limitations

One of the main limitations to this research is the sample imbalance for some cohorts. It could be argued that the group aged 50 or over provided a lower accuracy due to the lower sample spread, with the test sample number for that age group equaling 1518, compared to the other age groups with larger sample numbers. This shows a lack of balance in the test dataset and, to reduce this imbalance, new experiments were conducted with the same sample numbers among the groups. This was completed by randomly sampling 1518 samples from both the 26–49 group and the 18–25 group and recreating the graphs from the balanced dataset. The charts generated with the balanced data are shown alongside the original charts.

The relationship between class size (proportion of each demographic group of the overall population) and accuracy score can be investigated further to show the relationship between accuracy and proportion of the total population. Figure 8 shows a scatter plot which identifies that cohorts with higher numbers (and therefore a higher proportion of the total population) are more representative than cohorts of a smaller sample size, which show more extreme accuracy scores.

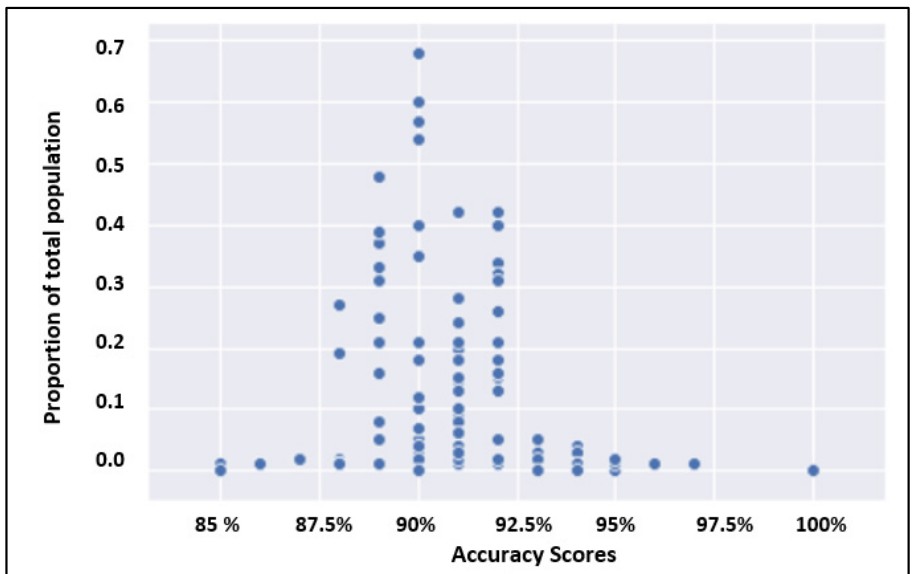

**Figure 8.** Scatter plot to show relationship between class size and accuracy score.

## 7. Conclusions

This paper focused on classifying happiness statements into one of seven different categories. Different demographic groups showed diverse accuracy scores produced by the algorithm. In determining the gender, marital status, parenthood status, and age, it was found that the results showed improved performance results for married and parent subgroups, and lower accuracies for the single and non-parent sub-groups. This may be due to the differing lengths of the happiness statements or different levels of complexity, and would benefit from further study. It should also be noted that, in order to improve this way of classifying happiness, different models may need to be trained on each demographic group.

Once a significant category of happiness has been established using the classifier, these insights could be used to make suggestions or recommendations to a user about an activity that could enhance their happiness based on events that have brought them happiness in the past. The user could then receive these suggestions or recommendations via a chatbot or other type of conversational agent. While conversational agents have been used in mental health care management in the past, it is necessary to understand the users in greater depth to make sure that appropriate recommendations are made based on the end user's demographic profile.

Other future work could involve using NLP and, in particular, POS (part-of-speech) tagging to identify the verbs in the happy statement that relate to an activity that has brought happiness. For example, words such as 'bought' and 'received' were identified in the texts. Further analysis with NLP could help identify what type of things were being bought or received that brought happiness. Future research could also involve investigating how PERMAH components—positive emotion, engagement, relationships, meaning, accomplishment, and health—could be used in the classification of texts in the HappyDB. Additional analysis could identify the types of words used in relation to the PERMAH components and how these may differ between gender, age, marital status and parenthood groupings.

**Author Contributions:** Conceptualization, C.S., E.E., M.M., S.O. and R.B.; methodology, formal analysis, writing—original draft preparation, C.S.; writing—review and editing, C.S., E.E., M.M. and R.B.; supervision, E.E., M.M. and R.B. All authors have read and agreed to the published version of the manuscript.

**Funding:** This research received no external funding.

**Institutional Review Board Statement:** Not applicable.

**Informed Consent Statement:** Not applicable.

**Data Availability Statement:** Data is contained within the article. A link to the dataset used in this research is at https://github.com/megagonlabs/HappyDB. (accessed on 5 May 2020).

**Conflicts of Interest:** The authors declare no conflict of interest.

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
