# Peer review of "How Machine Learning Classification Accuracy Changes in a Happiness Dataset with Different Demographic Groups"

_computers, doi:10.3390/computers11050083_

Round 1

Reviewer 1 Report

-The beginning of Abstract needs to be rewritten. It is not in the form of a scientific paper by now, so it needs a thorough revision. Please structure the abstract in a more compact and scientifically sound way, such that motivation, conclusions and experiments performed are well clear

-line 24-27 -> too general. Please rephrase and add more references. I think machine learning should be introduced in general, as a way of potentially being able to classify emotions, and then the rest should be introduced. Bring also examples where machine learning is applied in other fields, in order to then describe the applications to sentiment analysis more deeply. For example add: 

  1.          -Spiga, Ottavia, et al. "Machine learning application for patient stratification and phenotype/genotype investigation in a rare disease." Briefings in Bioinformatics 22.5 (2021): bbaa434.
  2.         -Bianchini, Monica, et al. "Deep neural networks for structured data." Computational Intelligence for Pattern Recognition. Springer, Cham, 2018. 29-51.
  3.        -Bengio, Yoshua, Andrea Lodi, and Antoine Prouvost. "Machine learning for combinatorial optimization: a methodological tour d’horizon." European Journal of Operational Research 290.2 (2021): 405-421.
  4.        -Dimitri, Giovanna Maria, et al. "Simultaneous transients of intracranial pressure and heart rate in traumatic brain injury: Methods of analysis." Intracranial Pressure & Neuromonitoring XVI. Springer, Cham, 2018. 147-151.
  5.       -Yang, Peng, and Yunfang Chen. "A survey on sentiment analysis by using machine learning methods." 2017 IEEE 2nd Information Technology, Networking, Electronic and Automation Control Conference (ITNEC). IEEE, 2017.
  6.      -Zhang, Lei, Shuai Wang, and Bing Liu. "Deep learning for sentiment analysis: A survey." Wiley Interdisciplinary Reviews: Data Mining and Knowledge Discovery 8.4 (2018): e1253.
  7.      -Maj, Carlo, et al. "Integration of machine learning methods to dissect genetically imputed transcriptomic profiles in alzheimer’s disease." Frontiers in genetics (2019): 726.

-line 30-35 -> rephrase as it is not scientifically sound. 

-line 34 -> "An algorithm like this": please avoid such kind of expressions, and rephrase 

-line 37: same for expressions like "Different models were trained to". Please rephrase a rewrite more scientifically sounding the whole introduction 

-add the references for lines 37-42

-describe more the work of reference [2]. Please add detailed description also in the intro from line 42

-in the related section I think it would be useful a table or a figure, explaining the structure of the database, to improve the readability of this section 

-rephrase line 121 (sentence "Definitions of the categories including examples can be seen in Table 1." meaning?)

-please add many more details to Table 1, explain the meaning and the relevance of it

-there is need to compute cross validation for the experiments results presented. A single hold-out trial is not sufficient. Please at least perform a 5 folds cross validation  and show the results

-Please restructure the classification evaluation section. There is no need of all of the subsections 2.3.1-2-3-4. Make them compact, and also present all of the equations in a more scientifically sounding way

-lines 191-193 -> they look out of place, please rephrase and make everything more smooth

-lines 195-206 -> some of these information are repeated, also it needs a complete rewriting to improve style and presentation of the contents. Moreover some sentences needs to be removed completely (Naïve Bayes -> a simple Bayesian classifier -> not appropriate for a scientific paper)

-lines 207-215 -> contents repeated and redundant. Please present the experimental results in a more appropriate way

-Table 3 improve consistently. Add Cross Validation and add a caption which is relevant and informative

-lines 217-220 -> it makes no sense, please improve  

-The whole experiments and discussion section needs to be improved. Add details, make sentences shorter. Even if results and study proposed are interested, in the way it is presented it is hardly readable

-Please add details and rewrite the conclusions section

Author Response

Comments and Suggestions for Authors (answers in blue below)

  1. -The beginning of Abstract needs to be rewritten. It is not in the form of a scientific paper by now, so it needs a thorough revision. Please structure the abstract in a more compact and scientifically sound way, such that motivation, conclusions and experiments performed are well clear.

Thank you for your comments. I have made changes to the first line in abstract. Included in abstract are:

Motivation  - to establish any variety in classification performance when tested on different demographic groups.

Conclusions - accuracy of prediction within this dataset declines with age, with the exception of the single parents’ subgroup.

Experiments - Using the happiness category field, we test different types of machine learning classifiers to predict what category of happiness the statements belong to, whether they indicate happiness relating to achievement or affection, for example. Tests were initially conducted with three distinct classifiers and the best performing model was the convolutional neural network (CNN).

  1. -line 24-27 -> too general. Please rephrase and add more references. I think machine learning should be introduced in general, as a way of potentially being able to classify emotions, and then the rest should be introduced. Bring also examples where machine learning is applied in other fields, in order to then describe the applications to sentiment analysis more deeply. For example add: 
  • Spiga, Ottavia, et al. "Machine learning application for patient stratification and phenotype/genotype investigation in a rare disease." Briefings in Bioinformatics5 (2021): bbaa434.
  • -Bianchini, Monica, et al. "Deep neural networks for structured data." Computational Intelligence for Pattern Recognition. Springer, Cham, 2018. 29-51.
  • -Bengio, Yoshua, Andrea Lodi, and Antoine Prouvost. "Machine learning for combinatorial optimization: a methodological tour d’horizon." European Journal of Operational Research2 (2021): 405-421.
  • -Dimitri, Giovanna Maria, et al. "Simultaneous transients of intracranial pressure and heart rate in traumatic brain injury: Methods of analysis." Intracranial Pressure & Neuromonitoring XVI. Springer, Cham, 2018. 147-151.
  • -Yang, Peng, and Yunfang Chen. "A survey on sentiment analysis by using machine learning methods." 2017 IEEE 2nd Information Technology, Networking, Electronic and Automation Control Conference (ITNEC). IEEE, 2017.
  • -Zhang, Lei, Shuai Wang, and Bing Liu. "Deep learning for sentiment analysis: A survey." Wiley Interdisciplinary Reviews: Data Mining and Knowledge Discovery4 (2018): e1253.
  • -Maj, Carlo, et al. "Integration of machine learning methods to dissect genetically imputed transcriptomic profiles in alzheimer’s disease." Frontiers in genetics(2019): 726.

Lines 24-27 are explaining how technology can be used in the area of positive psychology. This is a high level. Machine Learning is introduced in the next paragraph, and in more depth, in the ‘Related Works’ section. The articles that I have referenced are identified in the ‘Related Works’ section.

  1. -line 30-35 -> rephrase as it is not scientifically sound. 

I have reworded/re-phrased some of the text in this paragraph (see the ‘computers-1611377 - Reviewer 1 changes’ document).

  1. -line 34 -> "An algorithm like this": please avoid such kind of expressions, and rephrase 

As above (see the ‘computers-1611377 - Reviewer 1 changes’ document).

  1. -line 37: same for expressions like "Different models were trained to". Please rephrase a rewrite more scientifically sounding the whole introduction 

I have reworded/re-phrased some of the text in this paragraph (see the ‘computers-1611377 - Reviewer 1 changes’ document).

  1. -add the references for lines 37-42

There are no references necessary, as this is a description of the methodology used.

  1. -describe more the work of reference [2]. Please add detailed description also in the intro from line 42

Lines 30-33 of the ‘computers-1611377 - Reviewer 1 changes’ document have additional text relating to reference [2].

  1. -in the related section I think it would be useful a table or a figure, explaining the structure of the database, to improve the readability of this section

Table 1 shows a list of the categories of happy moments, which is the field that is predicted using the various machine learning classification systems. Would you like more detail relating to the demographic fields?

  1. -rephrase line 121 (sentence "Definitions of the categories including examples can be seen in Table 1." meaning?)

I have reworded/re-phrased some of the text in this paragraph (see 123- 124, in the ‘computers-1611377 - Reviewer 1 changes’ document).

  1. -please add many more details to Table 1, explain the meaning and the relevance of it

As above (see the ‘computers-1611377 - Reviewer 1 changes’ document).

  1. -there is need to compute cross validation for the experiments results presented. A single hold-out trial is not sufficient. Please at least perform a 5 folds cross validation  and show the results

Validation was performed on (20%) of the dataset using validation_split=0.2, for 5 epochs. Validation results are shown below:

Train on 64342 samples, validate on 16086 samples

Epoch 1/5

64342/64342 [==============================] - 490s 8ms/step - loss: 0.842 - acc: 0.7083 - val_loss: 0.4874 - val_acc: 0.8402

Epoch 00001: val_acc improved from inf to 0.84017, saving model to C:\Users\TeamKnowhow\Desktop\Research related\Code\weights_base.CovNet_GloVe.hdf5

Epoch 2/5

64342/64342 [==============================] - 439s 7ms/step - loss: 0.452 - acc: 0.8540 - val_loss: 0.3799 - val_acc: 0.8688

Epoch 00002: val_acc did not improve from 0.84017

Epoch 3/5

64342/64342 [==============================] - 447s 7ms/step - loss: 0.369 - acc: 0.8793 - val_loss: 0.3397 - val_acc: 0.8795

Epoch 00003: val_acc did not improve from 0.84017

Epoch 4/5

64342/64342 [==============================] - 437s 7ms/step - loss: 0.307 - acc: 0.8986 - val_loss: 0.3250 - val_acc: 0.8887

Epoch 00004: val_acc did not improve from 0.84017

Epoch 5/5

64342/64342 [==============================] - 439s 7ms/step - loss: 0.268 - acc: 0.9116 - val_loss: 0.2905 - val_acc: 0.8984

  1. -Please restructure the classification evaluation section. There is no need of all of the subsections 2.3.1-2-3-4. Make them compact, and also present all of the equations in a more scientifically sounding way

The subsections and associated numberings are in line with the MDPI formatting standards. The author does not see any way to present the equations in a more scientifically sounding way.

  1. -lines 191-193 -> they look out of place, please rephrase and make everything more smooth

I have moved these lines to the paragraph preceding Table 2, and reworded/re-phrased some of the text in this paragraph (see 187- 192, in the ‘computers-1611377 - Reviewer 1 changes’ document).

  1. -lines 195-206 -> some of these information are repeated, also it needs a complete rewriting to improve style and presentation of the contents. Moreover some sentences needs to be removed completely (Naïve Bayes -> a simple Bayesian classifier -> not appropriate for a scientific paper)

The first paragraph in the ‘Results’ section is reiterating what has already been mentioned in previous sections, as a way of summarizing what has already been introduced. Lines 207-214 have been reworded/re-phrased (in the ‘computers-1611377 - Reviewer 1 changes’ document).

  1. -lines 207-215 -> contents repeated and redundant. Please present the experimental results in a more appropriate way

I have reworded/re-phrased some of the text in this paragraph (see 207- 213, in the ‘computers-1611377 - Reviewer 1 changes’ document).

  1. -Table 3 improve consistently. Add Cross Validation and add a caption which is relevant and informative

See Point 11 for cross validation results. Would you like the cross validation results presented in this table?

  1. -lines 217-220 -> it makes no sense, please improve  

These lines explain that the CNN algorithm is producing the best performance and when we are looking at the complete test dataset (n=20,107), it produces an accuracy result of 90.83%, which is shown as the baseline in the following charts, as a black line, for comparative purposes.

  1. -The whole experiments and discussion section needs to be improved. Add details, make sentences shorter. Even if results and study proposed are interested, in the way it is presented it is hardly readable

I could show the graphs as a larger image size, for clarity of viewing, but the ease of comparative viewing will be removed.

  1. -Please add details and rewrite the conclusions section

The conclusion section gives a summary of the work undertaken, how the discoveries can be applied and any future considerations. No additional details are necessary?

Reviewer 2 Report

HappyDB and NLP were used to evaluate the models of classifying sentences of happiness. 

Major points 

BERT-based NLP models are available now. I recommend that authors compare word embeddings + ML/CNN with BERT-based models.

I recommend that authors split HappyDB dataset into train/val/test sets.

Image quality of Figure is low (low image resolution).

“to reduce this imbalance, new experiments were conducted with the same sample numbers among the groups.” Maybe, train set is also imbalance. If so, the effect of imbalance is not removed in Figure 2, 4, 6.

Table 4 may be biased because of the number of entries as shown by authors. This point should be more emphasized.

Minor points 

“2.2. Data science pipeline“ Please illustrate the pipeline as Figure.

Please clarify the hyperparameters of CNN model.

Please add columns for train/test sets to Table 2.

Author Response

Comments and Suggestions for Authors (responses in blue below)

HappyDB and NLP were used to evaluate the models of classifying sentences of happiness. 

Thank you for your comments. I have addressed the Major points highlighted by you, as below: 

  1. BERT-based NLP models are available now. I recommend that authors compare word embeddings + ML/CNN with BERT-based models.

I am aware of BERT-based NLP modelling, but I have applied the CNN model in this instance, where ease of deployment (1) was my main concern

1: Guo, W., Liu, X., Wang, S., Gao, H., Sankar, A., Yang, Z., Guo, Q., Zhang, L., Long, B., Chen, B.C. and Agarwal, D., 2020, October. Detext: A deep text ranking framework with bert. In Proceedings of the 29th ACM International Conference on Information & Knowledge Management (pp. 2509-2516).

  1. I recommend that authors split HappyDB dataset into train/val/test sets.

The data was split for validation (20%) using validation_split=0.2.

  1. Image quality of Figure is low (low image resolution).

The images are a certain size percentage ratio, to facilitate the presentation of two graphs on the same level. This is for comparison, where the reader can view the graphs generated using the original test sample dataset with the balanced test sample, for comparison. I can increase the size of the graphs, for clarity of viewing, but the ease of comparative viewing will be removed?

  1. “to reduce this imbalance, new experiments were conducted with the same sample numbers among the groups.” Maybe, train set is also imbalance. If so, the effect of imbalance is not removed in Figure 2, 4, 6.

The percentage split of the categories in the complete dataset compared to the train dataset can be viewed below. The same percentages per category in the complete dataset are reflected in the dataset used to train the model.

Category

Count - Total dataset

%

Count - Train

%

Affection

34,168

34%

27,372

34%

Achievement

33,993

34%

27,211

34%

Enjoy the Moment

11,144

11%

8,905

11%

Bonding

10,727

11%

8,609

11%

Leisure

7,458

7%

5,922

7%

Nature

1,843

2%

1,456

2%

Exercise

1,202

1%

953

1%

Total

100,535

80,428

  1. Table 4 may be biased because of the number of entries as shown by authors. This point should be more emphasized.

Table 4 simply shows the breakdown of all the results, for each demographic groups, for the complete dataset. This table includes the % accuracy for each demographic group, and the number in each cohort. 

I have addressed the Minor points highlighted by you, as below: 

  1. “2.2. Data science pipeline“ Please illustrate the pipeline as Figure.

Please find pipeline diagram attached. Do you think this should be added to the document?

  1. Please clarify the hyperparameters of CNN model.

Please find CNN Model summary, below:

Model: "sequential_3"

_________________________________________________________________

Layer (type)                 Output Shape              Param #  

=================================================================

embedding_3 (Embedding)      (None, 500, 100)          2331400  

_________________________________________________________________

dropout_7 (Dropout)          (None, 500, 100)          0         

_________________________________________________________________

conv1d_3 (Conv1D)            (None, 500, 64)           32064    

_________________________________________________________________

dropout_8 (Dropout)          (None, 500, 64)           0        

_________________________________________________________________

max_pooling1d_3 (MaxPooling1 (None, 250, 64)           0        

_________________________________________________________________

flatten_3 (Flatten)          (None, 16000)             0        

_________________________________________________________________

dense_5 (Dense)              (None, 100)               1600100  

_________________________________________________________________

dropout_9 (Dropout)          (None, 100)               0        

_________________________________________________________________

dense_6 (Dense)              (None, 7)                 707      

=================================================================

Total params: 3,964,271

Trainable params: 3,964,271

Non-trainable params: 0

  1. Please add columns for train/test sets to Table 2.

Please find revised Table 2 below, showing counts for total dataset, plus counts per category for the training data and test data:

Category

Count - Total dataset

Count - Train

Count - Test

Affection

34,168

27,372

6,796

Achievement

33,993

27,211

6,782

Enjoy the Moment

11,144

8,905

2,239

Bonding

10,727

8,609

2,118

Leisure

7,458

5,922

1,536

Nature

1,843

1,456

387

Exercise

1,202

953

249

Total

100,535

80,428

20,107

Reviewer 3 Report

The manuscript is well-written, while did the authors have the text pictures? if you do, please add such pictures to the manuscript.

Author Response

Hi. Can you clarify what you would like me to do, when you say "did the authors have the text pictures? if you do, please add such pictures to the manuscript", please? I'm not sure I understand what the text pictures are?

Reviewer 4 Report

This is the review report of the paper which is titled “How machine learning classification accuracy changes in a happiness dataset with different demographic groups “. 

The paper presents a good topic, however; the method novelty is poor which is the main concern. This work was already published in this paper 
"Understanding a happiness dataset: How the machine learning classification accuracy changes with different demographic groups"

https://ieeexplore.ieee.org/document/9631455

I don't see significant changes. results and dataset are the same with 34% similarity in content in terms of writing. 

I don't see it is worth publishing the paper. Please make real changes.

Author Response

Comments and Suggestions for Authors (responses in blue below)

  1. This is the review report of the paper which is titled “How machine learning classification accuracy changes in a happiness dataset with different demographic groups “. The paper presents a good topic, however; the method novelty is poor which is the main concern. This work was already published in this paper 
    "Understanding a happiness dataset: How the machine learning classification accuracy changes with different demographic groups"

https://ieeexplore.ieee.org/document/9631455

Thank you for your comments. This paper is an extended version of the original, which was presented at the ICTS4eHealth (IEEE International Conference on ICT Solutions for e-Health) workshop at the ISCC (IEEE Symposium on Computers and Communications (ISCC 2021) conference in September 2021.

  1. I don't see significant changes. results and dataset are the same with 34% similarity in content in terms of writing. 

Prior to this article’s submission for review, I have worked with the magazine editor to ensure that I have reworded the original sections of this paper, where they are now less than 30% similarity.

  1. I don't see it is worth publishing the paper. Please make real changes.

I have made some changes to this paper that were highlighted by the other reviewers.

Round 2

Reviewer 1 Report

The authors have made some changes to the paper, however some important major comments are still pending: 

  • I believe there is the need of improving all of the figures. At the current state the resolution is too low and they are not readable. 
  • the text is still full of some qualitative expressions that would need to be removed
  • The authors claim that formulas cannot be explained better. However in this form, they are not presented in a mathematical way, but with words and need to be written appropriately for a scientific paper. Equations presented in this way are not acceptable. use proper mathematical formulation 
  • For cross validation I intend the need of performing also some testing on a separate test set not used during the training/validation procedure. Keep out part of your dataset in turn and perform external testing one the external dataset and present then the performances tested on hold out datasets. 
  • Training should be extended for more epochs 
  • I still believe the related work section would benefit for further improvement, as well as the conclusion, and the comments of my first review still apply

Author Response

Please find answers to your review points below:

  1. I believe there is the need of improving all of the figures. At the current state the resolution is too low, and they are not readable. 

Thank you for this suggestion. I have enlarged the figures, as requested.

  1. The text is still full of some qualitative expressions that would need to be removed.

Thank you. We have now removed qualitative expressions in the text, and updated the text, where appropriate, to improve the general flow of the text. Thank you for pointing these out.

For this update to the document following your second review, I have made further changes in text. I have made updates in Lines 15-16, 34, 48, (related works - 57-58, 63, 65-66, 68-69, 78, 83-88, 93-94, 103-104, 109-113, 115), 420 and 435).  

  1. The authors claim that formulas cannot be explained better. However in this form, they are not presented in a mathematical way, but with words and need to be written appropriately for a scientific paper. Equations presented in this way are not acceptable. use proper mathematical formulation .

Thank you for explaining what was required. I have updated the document with the appropriate mathematical formulation.

  1. For cross validation I intend the need of also performing some testing on a separate test set not used during the training/validation procedure. Keep out part of your dataset in turn and perform external testing one the external dataset and present then the performances tested on hold out datasets. 

Thank you for your recommendation. Sorry that I may not have explained the validation process completely as part of the initial review. I will now try to show that the dataset is split into train/val/test sets or cross validation.

The data was initially split for testing (20% of 100,535 records) using the code below. The test dataset was printed out as an XLS file, which was plugged in as the test file further in the pipeline.

test=train.sample(frac=0.2)

This is where 80,428 records are used for training the model and 20,107 will be used for testing further on in the pipeline. Later on in the pipeline, the training data is further split (with code snippet below), where validation training is performed on 64,342 samples, and validation on 16,086 samples.

validation_split=0.2

  1. Training should be extended for more epochs. 

Thank you for this observation. When I started testing with the CNN model against the HappyDB dataset, I initially ran for 10 epochs, as the  optimal number of epochs to train most datasets is 10 or 11. I found that the accuracies started to level off around 4 – 5 and started to decrease after 5 or 6, so I set the number of epochs to 5, when fitting the model. I could have spent more time trying to improve the Accuracy of the CNN model by using a bigger pre-trained Glove model or using different loss function methods, apart from Cross-entropy, but for the purpose of this research, I wanted to choose 3 different classification methods, and using the best performer, then analyse the results per demographic spread.

  1. I still believe the related work section would benefit for further improvement, as well as the conclusion, and the comments of my first review still apply.

Thank you for your suggestion. I have made further changes in the text. Complete updates include change I made in the related works section – lines 57-58, 63, 65-66, 68-69, 78, 83-88, 93-94, 103-104, 109-113, 115, and lines 420 and 435 in the conclusion section. 

Reviewer 2 Report

>BERT-based NLP models are available now. I recommend that authors compare word embeddings + ML/CNN with BERT-based models.

>I am aware of BERT-based NLP modelling, but I have applied the CNN model in this instance, where ease of deployment (REF) was my main concern

Please clarify the lack of the comparison as limitation in the revision.  

>I recommend that authors split HappyDB dataset into train/val/test sets.

>The data was split for validation (20%) using validation_split=0.2.

Again, I recommend that authors split HappyDB dataset into train/val/test sets or use cross validation.

>I can increase the size of the graphs, for clarity of viewing, but the ease of comparative viewing will be removed?

I think that authors should increase the size of the graphs. Or, authors should change the size and types of font in the graphs.

>“2.2. Data science pipeline“ Please illustrate the pipeline as Figure.

>Please find pipeline diagram below. Let me know if you think this should be included in the revised document?

I think that this figure should be included in the paper.

>Please add columns for train/test sets to Table 2.

>Please find revised Table 2 below, showing counts for total dataset, plus counts per category for the training data and test data. Let me know if you think this should be included in the revised document?

I think that this revised tables should be included in the paper.

Author Response

Please find responses to your points below:

>BERT-based NLP models are available now. I recommend that authors compare word embeddings + ML/CNN with BERT-based models.

>I am aware of BERT-based NLP modelling, but I have applied the CNN model in this instance, where ease of deployment (REF) was my main concern

  1. Please clarify the lack of the comparison as limitation in the revision.  

To improve my knowledge of BERT-based NLP modelling, I have run through a Google Colab tutorial at the below URL, and it performs very well for the sentiment classification task. I have applied the CNN model in this instance, where ease of deployment was my main concern, and I was familiar with running these models. I could have spent more time trying to improve the accuracy by looking at other machine learning techniques, apart from CNN, or trying to improve the CNN model by using a bigger pre-trained Glove model or use different loss function methods, apart from Cross-entropy. But, for the purpose of this research, I wanted to choose 3 different classification methods, and using the best performer, then analyze the results per demographic spread. I will definitely use this type of modelling in future for other text classification tasks.

https://colab.research.google.com/github/jalammar/jalammar.github.io/blob/master/notebooks/bert/A_Visual_Notebook_to_Using_BERT_for_the_First_Time.ipynb#scrollTo=fvFvBLJV0Dkv

>I recommend that authors split HappyDB dataset into train/val/test sets.

>The data was split for validation (20%) using validation_split=0.2.

  1. Again, I recommend that authors split HappyDB dataset into train/val/test sets or use cross validation.

Thank you for your recommendation. Sorry that I may not have explained the validation process completely as part of the initial review. I will now try to show that the dataset is split into train/val/test sets or cross validation.

The data was initially split for testing (20% of 100,535 records) using the code below.

test=train.sample(frac=0.2)

This is where 80,428 records are used for training the model and 20,107 will be used for testing further on in the pipeline. Later on in the pipeline, the training data is further split (with code snippet below), where validation training is performed on 64,342 samples, and validation on 16,086 samples.

validation_split=0.2

>I can increase the size of the graphs, for clarity of viewing, but the ease of comparative viewing will be removed?

  1. I think that authors should increase the size of the graphs. Or, authors should change the size and types of font in the graphs.

Thank you for this suggestion. I have enlarged the figures, as requested.

>“2.2. Data science pipeline“ Please illustrate the pipeline as Figure.

>Please find pipeline diagram below. Let me know if you think this should be included in the revised document?

  1. I think that this figure should be included in the paper.

Thank you for this suggestion. I have added an additional diagram (Figure 1) for the pipeline of CNN model.

>Please add columns for train/test sets to Table 2.

>Please find revised Table 2 below, showing counts for total dataset, plus counts per category for the training data and test data. Let me know if you think this should be included in the revised document?

  1. I think that this revised tables should be included in the paper.

You are correct. The table would be enhanced by adding train/test breakdown for comparison. These additional columns have been added to Table 2.

Reviewer 4 Report

I don't see real improvement from the first version. 

Author Response

Thankyou. I have made further updates as per comments and recommendations of other reviewers. I hope it has improved the article.

Round 3

Reviewer 1 Report

The authors have improved the manuscript, however still some concerns holds

The whole cross validation procedure should also be reported in the paper. Moreover the accuracies for different train/test splitting should be reported, together with confidence levels. 

The epochs explanation should also be added, together with graphs showing the loss decrease as explained in the answer to my requests 

I still find the line graphs need improvements. All of the decimal numbers should be removed, and graphically they should be improved to be more readable

Table 4 is also not so clear. Should be re-organized as well as Figure 8

Author Response

Please find responses to your comments below (in blue):

The authors have improved the manuscript, however still some concerns hold:

  1. The whole cross validation procedure should also be reported in the paper. Moreover the accuracies for different train/test splitting should be reported, together with confidence levels. 

Thank you for your recommendation. I have explained the validation process completely in the document (lines 123 – 129). I have shown that the dataset is split into train/val/test sets for cross validation.

  1. The epochs explanation should also be added, together with graphs showing the loss decrease as explained in the answer to my requests.

Thank you for this observation. I have mentioned that 5 epochs were set at the validation step. I do not feel that the graphs showing the loss decrease as explained in one of my previous answers, would be too much details for this paper add may not add any additional value.

  1. I still find the line graphs need improvements. All of the decimal numbers should be removed, and graphically they should be improved to be more readable.

Thank you for your suggestion. I have removed the decimal numbers in all the charts, and I have made general enhancements to the graphs, to make them more readable.

  1. Table 4 is also not so clear. Should be re-organized as well as Figure 8.

Thank you for your recommendation. I have made changes to table 4: I have split the original table into 3 cohorts for ease of viewing (lines 311 – 320). This is to display the data in a clearer and more organized fashion. Similarly, I have made updates to Figure 8, changing the font to black, to make the figure sharper.

Reviewer 2 Report

“This is where 80,428 records are used for training the model and 20,107 will be used for testing further on in the pipeline. Later on in the pipeline, the training data is further split (with code snippet below), where validation training is performed on 64,342 samples, and validation on 16,086 samples.

validation_split=0.2”

This procedure is not described in the paper.

Author Response

Please find response to your comments below (in blue):

“This is where 80,428 records are used for training the model and 20,107 will be used for testing further on in the pipeline. Later on in the pipeline, the training data is further split (with code snippet below), where validation training is performed on 64,342 samples, and validation on 16,086 samples”.

validation_split=0.2”

  1. This procedure is not described in the paper.

Thank you for pointing this out to me. I have explained the validation process completely in the document (lines 123 – 129). I have shown that the dataset is split into train/val/test sets for cross validation.

Reviewer 4 Report

Dear Authors, 

I would expect to see real changes in the method otherwise it is just like your previous work. 

Author Response

Thank you. I have responded to your comment below (in blue): 

Dear Authors, 

  1. I would expect to see real changes in the method otherwise it is just like your previous work.

Thank-you. I have made further updates to the methodology section, as per comments and recommendations of other reviewers.

Round 4

Reviewer 1 Report

I thank the author for having addressed my concerns

Author Response

Thank you for your comment. I am happy I have addressed your concerns. 

Reviewer 2 Report

No comments.

Author Response

Thank you for working with me through this process of improving my paper.

Reviewer 4 Report

The problem with this paper is that doesn't have any new contributions on top of previous authors' work 

https://ieeexplore.ieee.org/document/9631455

Please change the method, improve the results, show new contributions, then send it to a journal. 

I don't find it interesting to add little bit on previous work and submit it. 

Author Response

Thank you for your comments and suggestion of how to improve my paper. I have taken your recommendations on board.

This paper is an extended and improved version of the original paper, which was presented at the ICTS4eHealth (IEEE International Conference on ICT Solutions for e-Health) workshop at the ISCC (IEEE Symposium on Computers and Communications (ISCC 2021) conference in September 2021.

The main contributions of this extended paper include revising the methods with additional analysis on the balanced data, where new experiments were undertaken across three age groups and the original results were improved, where compared with the new balanced samples using new line charts. This was due to one of the age cohorts being notably smaller than the other 2 age groups.

The other contribution involves new assessments being conducted regarding text length and complexity, to try to explain the patterns observed. The Flesch Kincaid Grade Level readability formula was used to calculate the complexity of the text and mean length (per happiness statement) was used to determine the length of texts for different age groups, for the single parents only group, to compare with the other demographic groups.

This work introduces an experiment that was carried out to ascertain if trained machine learning classifiers (using happiness statements) were able to classify unlabeled happiness statements into one of seven different categories.

The main contributions of this extended paper include additional analysis on the balanced data  (see section 2.5 Test dataset imbalance) within the methods section, and new assessments conducted regarding text length and complexity, to try to explain the patterns observed. The Flesch Kincaid Grade Level readability formula was used to calculate the complexity of the text and mean length (per happiness statement) was used to determine the length of texts for different age groups, for the single parents only group, to compare with the other demographic groups.

Even though the complexity scores calculated were inconclusive, and the median text length analysis is not reflecting what is expected (i.e., an increase in text length should improve the classification task) the author feels that it has been important to use these new methods to try to explain why the results were showing lower accuracies for the single parent sub-groups.

Round 5

Reviewer 1 Report

I confirm the impression of last revision

Reviewer 4 Report

Sorry, I don't see it is worth publishing it as a journal paper